# Targeting Vesicular LGALS3BP by an Antibody-Drug Conjugate as Novel Therapeutic Strategy for Neuroblastoma

**DOI:** 10.3390/cancers12102989

**Published:** 2020-10-15

**Authors:** Emily Capone, Alessia Lamolinara, Fabio Pastorino, Roberta Gentile, Sara Ponziani, Giulia Di Vittorio, Daniela D’Agostino, Sandra Bibbò, Cosmo Rossi, Enza Piccolo, Valentina Iacobelli, Rossano Lattanzio, Valeria Panella, Michele Sallese, Vincenzo De Laurenzi, Francesco Giansanti, Arturo Sala, Manuela Iezzi, Mirco Ponzoni, Rodolfo Ippoliti, Stefano Iacobelli, Gianluca Sala

**Affiliations:** 1Department of Medical, Oral and Biotechnological Sciences, University of Chieti-Pescara, 66100 Chieti, Italy; emily.capone@unich.it (E.C.); daniela.dagostino@unich.it (D.D.); sandra.bibbo@unich.it (S.B.); rossano.lattanzio@unich.it (R.L.); vpanella@unite.it (V.P.); michele.sallese@unich.it (M.S.); delaurenzi@unich.it (V.D.L.); 2Center for Advanced Studies and Technology (CAST), Via Polacchi 11, 66100 Chieti, Italy; alessia.lamolinara@unich.it (A.L.); c.rossi@unich.it (C.R.); m.iezzi@unich.it (M.I.); 3Department of Medicine and Aging, University of Chieti-Pescara, 66100 Chieti, Italy; 4Laboratory of Experimental Therapies in Oncology, IRCCS Istituto Giannina Gaslini, 16147 Genoa, Italy; fabiopastorino@gaslini.org (F.P.); mircoponzoni@gaslini.org (M.P.); 5MediaPharma S.r.l., Via della Colonnetta 50/A, 66100 Chieti, Italy; r.gentile@mediapharma.it (R.G.); g.divittorio@mediapharma.it (G.D.V.); enza.piccolo@iisdettafermi.it (E.P.); 6Department of Life, Health and Environmental Sciences, University of L’Aquila, 67100 Coppito, Italy; s.ponziani@mediapharma.it (S.P.); francesco.giansanti@cc.univaq.it (F.G.); rodolfo.ippoliti@univaq.it (R.I.); 7Department of Gynecology and Obstetrics, Catholic University, 00168 Rome, Italy; vale.iacobelli@gmail.com; 8Laboratory of Molecular Genetics, Department of Psychological, Health and Territorial Sciences, School of Medicine and Health Sciences, G. d’Annunzio University, 66100 Chieti-Pescara, Italy; arturo.sala@brunel.ac.uk; 9Centre for Inflammation Research and Translational Medicine (CIRTM), Brunel University London, Uxbridge UB8 3PH, UK

**Keywords:** Antibody-Drug Conjugates (ADC)s, LGALS3BP, neuroblastoma, targeted therapy

## Abstract

**Simple Summary:**

Antibody Drug Conjugates are an emerging class of biopharmaceuticals that have seen an impressive increase of attention in the field of cancer therapy. Here, we describe the therapeutic activity of 1959-sss/DM3, a non-internalizing ADC targeting LGALS3BP, a secreted, extracellular vesicles-associated protein expressed by the majority of human cancers, including neuroblastoma. We show that 1959-sss/DM3 treatment can cure mice with established neuroblastoma tumours in pseudometastatic, orthotopic and Patient Derived Xenograft models.

**Abstract:**

Neuroblastoma is the most common extra-cranial solid tumor in infants and children, which accounts for approximately 15% of all cancer-related deaths in the pediatric population. New therapeutic modalities are urgently needed. Antibody-Drug Conjugates (ADC)s-based therapy has been proposed as potential strategy to treat this pediatric malignancy. LGALS3BP is a highly glycosylated protein involved in tumor growth and progression. Studies have shown that LGALS3BP is enriched in extracellular vesicles (EV)s derived by most neuroblastoma cells, where it plays a critical role in preparing a favorable tumor microenvironment (TME) through direct cross talk between cancer and stroma cells. Here, we describe the development of a non-internalizing LGALS3BP ADC, named 1959-sss/DM3, which selectively targets LGALS3BP expressing neuroblastoma. 1959-sss/DM3 mediated potent therapeutic activity in different types of neuroblastoma models. Notably, we found that treatments were well tolerated at efficacious doses that were fully curative. These results offer preclinical proof-of-concept for an ADC targeting exosomal LGALS3BP approach for neuroblastomas.

## 1. Introduction

Metastatic neuroblastoma is a childhood cancer that kills the majority of the affected children, after emergence of drug resistance. Standard treatment, including chemotherapy, radiation therapy and stem cell transplant are of limited efficacy and cause important short and long-term side effects [1,2]. Although a significant improvement of the outcome has been achieved with the introduction of hu14.18K322A (dinutuximab), an antibody against the surface antigen GD2 [3,4], a significant number of patients fail to respond to this targeted agent and undergo severe uncontrollable neuropathic pain as a main side effect [5]. Thus, development of further therapeutics for this aggressive form of childhood cancer is urgently needed. 

LGALS3BP (Galectin- 3 binding protein, aka Mac-2 BP or 90K), is a human secreted protein expressed by the large majority of human cancers, while being virtually undetectable or expressed at low levels in normal human tissues [6,7,8,9]. Whereas the bio-physiological role of LGALS3BP is not yet fully understood, accumulating evidence has shown that the protein may be involved in cancer growth and progression [10]. Notably, significantly elevated expression of LGALS3BP in the serum or tumor tissue has been found to be associated with poor clinical outcome in patients with a variety of malignancies, including neuroblastoma [8,9,11,12,13,14,15,16]. Recent studies have revealed that LGALS3BP plays a key role in the cross-talk between neuroblastoma cells and the tumor microenvironment (TME) [17]. Indeed, LGALS3BP is one of the most abundant surface component of cancer-derived EVs [18,19,20]. EVs have a prominent role in inducing a TME that is favorable to cancer progression through the induction of immune escape, stimulation of angiogenesis and preparation of the pre-metastatic niche [21,22]. Of particular interest is the observation that neuroblastoma-derived EVs are incorporated by mesenchymal stromal cells (MSC)s, where they induce, in a LGALS3BP-dependent manner, the secretion of pro-tumorigenic cytokines and chemokines, including IL-6 and IL-8/CXCL8 [23], providing an opportunity for immune targeting strategies.

Antibody-Drug Conjugates (ADC)s are an emerging class of biopharmaceuticals for cancer therapy [24,25]. Classically, monoclonal antibodies (mAb) with high internalization capacity have been used in ADC development, as they mediate efficient delivery of the conjugated toxins inside target tumor cells. However, recent studies have revealed that ADCs can also be generated using non-internalizing antibodies [26], targeting the tumor or its stroma [27,28,29]. In the present study, we investigated the therapeutic potential of an ADC targeting exosomal LGALS3BP, consisting of 1959-sss, an engineered humanized anti-LGALS3BP antibody, where the residual cysteines of the light chains was directly coupled to the maytansinoid SH-DM3, through formation of a disulfide bridge [30]. DM3 is a chemical derivative of maytansine belonging to the tubulin- binding ADC payload class with a cell killing potency in the picomolar range [31].

1959-sss/DM3 ADC showed potent and specific therapeutic activity in multiple models of LGALS3BP expressing human neuroblastoma. The data presented here, strongly encourage further development of this ADC targeting exosomal LGALS3BP in neuroblastoma. 

## 2. Results

### 2.1. LGALS3BP is Expressed at the Surface of Neuroblastoma Derived EVs

We analyzed LGALS3BP expression at mRNA and protein level in a panel of neuroblastoma cell lines. In agreement with previous reports [17,32], LGALS3BP was expressed in neuroblastoma cell lines, independently of MYCN amplification. Interestingly, hNB, a cell line established in our laboratory from a 4-yr old children with metastatic neuroblastoma [33] did not express LGALS3BP neither at mRNA or protein level (Figure 1A–C). Previously, LGALS3BP was reported to be one of the most abundant protein of tumor released exosomes/EVs, where it localizes at the surfaceoma [11]. Using confocal imaging, we confirmed that LGALS3BP co-localizes with the exosome markers CD-63 and CD-81 (Figure 1D and Appendix A) and its mature form is enriched in exosomes/EVs isolated from SKNAS but not hNB neuroblastoma cells (Figure 1E and Appendix A), providing an opportunity for specific targeting with 1959 antibody (Figure 1F).

### 2.2. 1959-sss/DM3 Exhibits Therapeutic Activity against Multiple Neuroblastoma Models in a Target- Dependent Manner

To evaluate the therapeutic efficacy of 1959-sss/DM3 in neuroblastoma, we tested the cytotoxic activity of SH-DM3 in a panel of human neuroblastoma cell lines. An efficient killing activity was seen in all tested cell lines, indicating that the selected neuroblastoma cells are sensitive to SH-DM3 (Appendix A). Moreover, the maytansine derivative showed potent cell killing activity in macrophages (Appendix A) both unpolarized (M0) and stimulated with conditioned medium obtained from neuroblastoma cell line SKNAS, as previously reported [34]. Initially, the therapeutic activity of 1959-sss/DM3 was evaluated in a subcutaneous neuroblastoma xenograft model, SKNAS. For these experiments, control vector-infected (shCTR) and LGALS3BP-knockdown (shLGALS3BP) cells were implanted subcutaneously into nude mice and when tumors reached an average volume of around 100mm3, animals were randomized to treatment by intravenous injections with control vehicle (PBS) or 1959-sss/DM3 twice a week for two consecutive weeks at the dose of 10 mg/kg. The administration of 1959-sss/DM3 caused a complete remission of tumors originating from shCTR cells (Figure 2A), which was accompanied by a significant prolonged survival of treated animals (Figure 2B). The therapeutic activity was long-lasting, as no tumor re-growth was observed in any of the treated animals up to 80 days from the start of ADC injections (Figure 2B). In contrast, after a transitory reduction of tumor volume, mice implanted with shLGALS3BP cells did not show any significant response to ADC treatment (Figure 2A,B). The transient decrease in volume observed in tumors originating from shLGALS3BP cells was probably due to the limited efficiency of shRNA silencing, which resulted in only about 50% reduction of target expression relative to cells infected with the control retrovirus (Appendix AC). To further study the specificity of 1959-sss/DM3, we examined the therapeutic activity of the ADC against hNB cells, which are completely target negative. Indeed, no significant therapeutic activity of 1959-sss/DM3 was observed in mice xenotransplanted with hNB cells (Appendix A). In line with this observation, staining of live tumor cells with the humanized 1959 anti-LGALS3BP antibody, produced a characteristic punctate pattern in proximity of the external surface of SKNAS, as previously observed in LGALS3BP^+^ neuroblastoma cells [30]. In contrast, no staining was detected in hNB cells (Figure 2C). Similarly, immunohistochemical analysis revealed that SKNAS, but not hNB tumor xenografts, were positive to LGALS3BP (Figure 2D). Ex-vivo immunofluorescence analysis of tumor masses explanted from mice treated for 72 h with the 1959-sss/DM3 showed antibody accumulation in cancer tissues of SKNAS, but not hNB, injected animals (Figure 2E). All together, these results strongly indicate target-dependent antibody accumulation.

To assess the therapeutic activity of 1959-sss/DM3 in a more physiological setting, SH-SY5Y-LUC cells were orthotopically implanted in the adrenal glands of nude mice, and animals were treated with the 1959-sss/DM3 using the same dosage schedule implemented in previous experiments. Also in this model, we observed complete inhibition of tumor growth (Figure 3A,B), and a significant extension of mice survival (Figure 3C). To confirm that the anticancer effect of the ADC was due to the release of the SH-DM3 payload, and not to antibody-dependent cellular cytotoxicity (ADCC) or complement mediated tumor cell lysis, nude mice harboring subcutaneous xenografts of SH-SY5Y cells were treated with 1959-sss only. As shown, the naked antibody was unable to reduce or delay tumor growth, proving that the anticancer activity of the ADC was caused by the payload (Appendix AA). 1959-sss/DM3 was well tolerated in mice, no body weight loss was observed during treatment nor biochemical alterations were found in blood samples collected in animals after the last treatment (Appendix AB).

Next, we evaluated the therapeutic activity of 1959-sss/DM3 against Patient-Derived Xenografts (PDX)s of neuroblastoma. Three highly aggressive, stage IV, MYCN amplified PDXs, COG-N-636, COG-N-603 and COG-N-453, were used. COG-N-636 and COG-N-603, which overall showed the higher expression levels of ADC target were selected for the in vivo studies (Figure 4A,B). When tumors reached an average volume of 100–200 mm^3^, animals were randomized to treatment by intravenous injections with control vehicle (PBS) or 1959-sss/DM3 twice a week for two consecutive weeks at the dose of 10 mg/kg. Treatments with the 1959-sss/DM3 ADC caused complete eradication of tumours in the COG-N-636 PDX model, whereas 5 out of 6 mice injected with COG-N-603 PDX cells were tumour-free at the time of manuscript submission, with only one animal showing a partial response (Figure 4C). The therapeutic activity was long-lasting, since tumour regrowth was only observed in one of the COG-N-603 PDX transplanted mice after 70 days from antibody injections. None of the COG-N-636 PDX transplanted mice showed signs of tumour regrowth after 150 days since the start of treatments.

### 2.3. 1959-sss/DM3 Strongly Inhibits Neuroblastoma Metastasis Formation

Neuroblastoma is a highly metastatic disease, therefore we wanted to assess the efficacy of 1959-sss/DM3 in a metastatic model. For these assays, we used three cell lines expressing different levels of the target, SKNAS (LGALS3BP high), Kelly (LGALS3BP low) and hNB (LGALS3BP negative). Spreading efficiency (i.e. % of animals producing metastatic lesions after intravenous cell injection) and timing (number of days after injection needed for ex-vivo detection of metastatic lesions by H&E staining) were established to select the appropriate starting points for treatment, as described [35,36]. A schematic representation of the experimental setting is shown in Figure 5A. Depending on the cell line used, metastases were detected in lungs, kidneys, liver and bone marrow. Maximal inhibition (90% reduction) of SKNAS metastasis formation was observed in liver (the organ for which SKNAS cells show preferential tropism) followed by lungs (80% reduction) and kidneys (60% reduction) (Figure 5B). Importantly, animal treated with equimolar doses of free SH-DM3 showed no inhibition of metastases formation, indicating that drug accumulation in tumor tissues and the local release of the payload are keys for the therapeutic effect of the ADC (Figure 5B). A significant response was also observed in mice injected with Kelly cells, which express low levels of the target but are highly sensitive to the payload (Appendix AA). Treatment with 1959-sss/DM3 caused 60–70% inhibition of metastasis formation in liver, while completely abolished dissemination in the bone marrow (an organ typically colonized by Kelly, but not SKNAS or hNB cells), indicating that this treatment may be used to combat minimal residual disease (Figure 5C). Finally, the hNB model was used to verify the target dependency of the ADC in this pseudo-metastatic setting. In preliminary experiments, the highly metastatic hNB cells displayed a strong tropism for the lungs, with no metastatic lesions observed in liver, kidneys or bone marrow. We then compared the growth of SKNAS (LGALS3BP highly positive) and hNB (LGALS3BP negative) metastases in mice treated with 1959-sss/DM3.

A strong reduction (about 80%) of lung metastatic lesions was observed in mice injected with SKNAS cells, whereas hNB-derived metastases developed normally in the presence of the therapeutic antibody (Figure 5D, left panels). Immunohistochemical analysis showed high expression of LGALS3BP in SKNAS-derived lung metastases, but no expression in those derived by hNB (Figure 5D, right panels), further confirming target dependence of the ADC’s therapeutic effect.

## 3. Discussion

Neuroblastoma is a childhood cancer and one of the major causes of oncological death in infancy. Metastatic neuroblastoma shows initial response to therapeutic interventions, but typically relapses into an incurable form of the disease [37,38]. Children with neuroblastoma are still treated with toxic cocktails of chemotherapeutic drugs and radiation therapy. Molecularly targeted approaches would be very appealing since they could lead to less toxicity and better clinical outcomes. However, the main neuroblastoma oncogene, MYCN, is still undruggable, and small molecule inhibitors targeting kinases activated in neuroblastoma, such PI3K or first generation ALK inhibitors, are either too toxic or made futile by the development of drug resistance, requiring the continuous search for more effective inhibitors [37,38]. Although, encouraging results were recently obtained with second and third generation ALK inhibitors which showed to overcome resistance and demonstrated significant therapeutical efficacy in several studies [39,40,41,42]. Immunotherapy with monoclonal antibodies is very relevant in the context of paediatric oncology because it does not necessarily rely on the presence of specific mutations. Indeed, the most important recent therapeutic breakthrough in neuroblastoma has been the development of the GD2 antibody (dinutuximab). Dinutuximab is a human/mouse chimeric, naked antibody recognising the disialoganglioside GD2, present on the surface of all neuroblastoma cells, which prolongs the life of children with relapsed, metastatic neuroblastomas [43]. However, only about half of the children respond positively to the antibody therapy, and the treatment causes important side effects, such as severe pain. A novel immunotherapy approach, based on conjugated antibodies (ADC)s has emerged as promising strategy for the development of efficient therapy against neuroblastoma. Indeed, two new ADCs have been recently deployed to target ALK and GPC2 expressed on the surface of neuroblastoma cells [44,45]. The mechanism of action of the ADC relies on the internalization of the antibodies that are linked to potent cytotoxic molecules. Initially, only tumor associated antigens or tumor growth factor receptors expressed on the surface of cancer cells were considered as credible targets for ADCs, but, more recently, also stromal components of the tumour microenvironment have been explored as potentially actionable targets [46,47]. In principle, using appropriate combinations of antibodies, linkers and payloads, ADCs that targeted the tumor milieu could improve the therapeutic efficacy by causing the destruction of both tumor cells and pro-tumorigenic macrophages, CAFs (Cancer-associated fibroblasts) and neo-endothelial cells (pericytes). We have recently developed a new non-internalizing ADC, called 1959-sss/DM3, which is endowed with a marked therapeutic activity in mouse models of human melanoma. The disulfide-linked payload is released within the reducing tumour microenvironment, therefore a therapeutic effect could be generated without antibody internalization [30]. We and others have previously ascertained that LGALS3BP is expressed in tumor cells and the extracellular matrix in the majority of neuroblastoma biopsies, but it is not accumulated in adjacent normal tissues. Furthermore, engagement of the LGALS3BP receptor galectin 3 mediates IL-6 production from bone marrow mesenchymal stem cells. Accordingly, it has been recently observed that serum levels of LGALS3BP are increased compared to controls in patients with neuroblastoma [17,23,30,32]. In this study, we confirmed that LGALS3BP is expressed by the vast majority of classical and patient-derived neuroblastoma cell lines. We detected the fully mature and secreted form of the protein in neuroblastoma produced EVs that can be recognized by the 1959 therapeutic antibody. Importantly, we have unequivocally proven that the ADC accumulates in neuroblastoma tumor tissues in a target dependent manner (Figure 2). We showed that 1959-sss/DM3 antibody induced significant shrinkage of established subcutaneous and orthotopic neuroblastomas, in a target-dependent manner. Moreover the ADC also inhibited metastatic dissemination in the liver, lungs and bone marrow of NSG mice injected intravenously with multiple neuroblastoma cell lines. Most importantly, the therapeutic ADC induced complete eradication of subcutaneous tumour masses, considerably prolonging survival, of mice transplanted with patient-derived high-risk neuroblastomas with MYCN amplification. One could speculate that the therapeutic effect is achieved using a mechanism similar to that described for the clinical approved anti-CD30 SGN-35 [48]. It has been proposed that CD30+ exosomes released from malignant cells are incorporated by cells of the tumor microenvironment, increasing local antigen density and enhancing the toxic activity of the ADC [49]. It is important to underline that neuroblastoma, similar to other cancers, is a very heterogeneous tumor, and therefore the different levels of expression of the target in the patient population (as observed in our cell lines panel and PDX setting) could limit this therapeutic approach to a selected subset of neuroblastoma patients.

In terms of safety, we would like to highlight that the 1959-sss/DM3 was well tolerated in mice (Appendix A) and rabbits, the latter being the only species among 15 examined where the 1959-sss antibody cross-reacts, as previously described [30]. Moreover, therapeutic studies using a rabbit tumor model [50] should be performed to predict whether the serum level of LGALS3BP may affect the efficacy of this ADC due to a potential sink effect. 

In conclusion, the results presented in this study validate LGALS3BP as a suitable target for ADC therapy and provides a rationale for further clinical development of an anti-LGALS3BP based ADC for the treatment of neuroblastomas.

## 4. Methods

### 4.1. Cell Lines and Biochemicals

Melanoma (A375m), neuroblastoma (SKNAS, SH-SY5Y, SHEP, IMR32) and murine RAW264.7 macrophages were purchased from American Type Culture Collection (Rockville, MD, USA). Neuroblastoma LAN-5 cell line was purchased from Leibniz Institute DMSZ (German Collection of Microorganism and Cell Culture GmbH). Neuroblastoma Kelly cell line was purchased from Sigma-Aldrich (St. Louis, MO, USA). The primary human neuroblastoma HNB cell line was isolated from a non-MYCN amplified tumour metastasized in the neck of a 3-year-old male patient. The 11q and ALK status is not known. [33]. All cell lines were cultured less than 3 months after resuscitation. The cells were cultured according to manufacturer’s instructions, using a medium supplemented with 10% heat-inactivated fetal bovine serum (FBS; Invitrogen, Carlsbad, CA, USA), l-glutamine, 100 units/mL penicillin, and 100 μg/mL streptomycin (Sigma-Aldrich Corporation, St. Louis, MO, USA), and incubated at 37 °C in humidified air with 5% CO_2_. 

SH-SY5Y cells were infected with retrovirus expressing the firefly luciferase (luc) gene, as previously reported [51]. Luciferase activity of retrovirally transduced cells was confirmed by bio-luminescent imaging (BLI, Lumina-II, Caliper Life Sciences, Hopkinton, MA, USA) after a 10 min incubation with 150 μg/mL d-luciferin (Caliper Life Sciences) and diluted in tissue culture medium as previously described [51].

All cell lines were tested for mycoplasma contamination, authenticated at time of experimentation by multiplex STR-profiling test (PowerPlex^®^ Fusion, Promega, Milan, Italy) by BMR Genomics (Padova, Italy) and validated using ATCC STR, DSMZ STR and NCBI databases. 

For PDX models, cell line stocks were obtained from the Children’s Oncology Group (COG) Cell Culture and Xenograft Repository at Texas Tech University Health Sciences Center (www.COGcell.org). Patient derived cell lines were cultured in vitro using IMDM medium supplemented with 20% heat-inactivated fetal bovine serum (FBS; Invitrogen), 1% ITS (Insulin-Tranferrin-Selenium; Thermo Fisher Scientific, Waltham, MA, USA) 100 units/mL penicillin, and 100 μg/mL streptomycin (Sigma-Aldrich Corporation, St. Louis, MO, USA), and incubated at 37 °C in humidified air with 5% CO_2_. For LGALS3BP stable knockdown, 21-nucleotide sequence corresponding to nucleotide 2216–2236 of human LGALS3BP mRNA (NCBI Accession NM-005567.3) or a 21-nucleotide sequence with no significant homology to any mammalian gene sequence serving as a non-silencing control (OligoEngine, Hercules, CA, USA) were inserted into the pSUPER.retro.puro (OligoEngine, Seattle, WA, USA) The generation of SKNAS knock-down cells was performed according to the methods described in our previous report [52].

### 4.2. Cytotoxicity Assays

Cell proliferation was assessed by MTT [3-(4,5-dimethyldiazol-2-yl)-2,5-diphenyl tetrazolium bromide] assay (Sigma-Aldrich). Cell lines were seeded into 24-well plates at a density ranging from 2 × 10^3^ to 5 × 10^3^ cells/well in 500 μL of complete culture medium, cells were treated with free SH-DM3 at indicated concentration in triplicates and further incubated for 72 h. At the end of the incubation period, cells were incubated with 200 μL of MTT solution (medium serum free with 0.5 mg/mL of MTT) for further 2 h. After removal of MTT solution, 200 μL of dimethyl sulfoxide (DMSO) was added to the wells for 10 min and the absorption value at 570 nm was measured using a multi-plate reader. All experiments were performed in triplicate.

### 4.3. 1959-sss/DM3 Purification and DAR Calculation

The 1959-sss/DM3 ADC was obtained as described [30]. All ADC products were analyzed by SDS-PAGE and size exclusion chromatography (Superdex200 10/300GL; GE Healthcare, Chicago, IL, USA).

All ADC batch after purification, were analyzed by HIC or LC-MS for Drug-Antibody Ratio (DAR) determination, as previously described [30].

### 4.4. Animal Studies

#### 4.4.1. Xenograft Experiments

Homozygous CD1 nu/nu athymic female mice (4–6-week old) were purchased from Charles River Laboratories, Milan, Italy and maintained at 22–24 °C under pathogen-limiting conditions as required. Cages, bedding, and food were autoclaved before use. Mice were given a standard diet and water ad libitum and acclimatized for 2 weeks before start of the experiments. Housing and all procedures involving the mice were performed according to the protocol approved by the Institutional Animal Care and Use Committee (Authorization n° 629/2015-PR).

For subcutaneous tumor growth, five million of exponentially growing SKNAS cells (shCTR and shLGALS3BP) and SH-SY5Y cells or two million of hNB cells were implanted s.c. in PBS into the right flank of the mice. For PDXs assays, we obtained material and expanded PDX from mouse to mouse through surgical implantation. In detail, tumor was cut with sterile scalpel into fragments of 2-3 mm^3^, which were implanted with a drop of Matrigel into subcutaneous pocket of anesthetized mouse; tumor growth was monitored weekly. F2 generation tumors of approximately 100 mm^3^ were used for the therapeutic experiments. For all experiments, animals were randomly divided and intravenously injected with 1959-sss/DM3 (10 mg/kg) or naked 1959-sss (10 mg/kg) or vehicle (PBS). Doses and schedules are described in the individual figure legends. Tumor volume was monitored weekly by a caliper and calculated using the following formula: tumor volume (mm^3^) = (length × width^2^)/2. A tumor volume of 1.5 cm^3^ was chosen as endpoint for all experiments after which mice were sacrificed and tumors dissected, fixed with formalin and embedded in paraffin. 

#### 4.4.2. Experimental Metastasis Assay 

NSG mice were purchased from Jackson Laboratory and bred in the animal facility of CeSI-Met, G. D’Annunzio University, Chieti. Animal care and experimental procedures were approved by the Ethics Committee for Animal Experimentation of the Institute according to Italian law (Authorization n° 292/2017-PR). 8-weeks old female NSG mice were injected via the lateral tail vein with 5 × 10^5^ Kelly or 1 × 10^6^ SKNAS or 2 × 10^4^ hNB neuroblastoma cells; after two weeks (for KELLY and SKNAS cell lines) and one week (for hNB cell line), mice were randomly divided into two groups that received vehicle (PBS) or 1959-sss/DM3 (10 mg/kg) or free SH-DM3 (0.1 mg/kg) every three days for three treatments. The animal health status was monitored daily and body weight was measured once a week during the treatments. 3 days after last treatment (18 days from hNB cell injection and 25 days from KELLY and SKNAS cell injections) mice were sacrificed and organs were harvested, fixed in 10% neutral buffered formalin, paraffin embedded, sectioned and stained with Hematoxylin and Eosin for metastasis analysis. To optimize the detection of microscopic metastases and ensure systematic uniform and random sampling, lungs and livers were cut transversally into 2.0 mm thick parallel slabs with a random position of the first cut in first 2mm of the organ, resulting in 5–8 slabs for lungs and 6–8 slabs for livers. The slabs were then embedded cut surface down and sections were stained with Haematoxylin and Eosin. Slides were independently evaluated by two pathologists to quantify the number of tumor lesions in the organs harvested. The major leg bones were harvested for extraction of bone marrow cells, by cutting the edges of the bones and flushing with 1 mL syringes containing PBS through the bone; the cells extracted were stained with primary anti-GD2 antibody (Clone 14.G2a, Abcam, Cambridge, United Kingdom), following by AlexaFluor-488 anti-mouse IgG (A11017, Molecular Probes, Life Technologies, Carlsbad, CA, USA), as secondary antibody for cytofluorimetric analysis of neuroblastoma GD2+ cells. Lung metastatic area was evaluated on five digital images of each sample (X10 microscopic fields) using Adobe Photoshop by selecting each metastatic area with the lasso tool and reporting the total number of pixels indicated in the histogram window as percentage of the total lung area in pixel. 

#### 4.4.3. Orthotopic Experiments 

Female athymic Nude-Foxn1*^nu^* mice were purchased from Envigo (Envigo, Bresso, Italy) and housed under pathogen-free conditions. Experiments were approved by ethical committee of the Italian Ministry of Health (n.: 661/2016-PR), in compliance with the “ARRIVE” guidelines (Animals Research: Reporting in Vivo Experiments). Five-week-old mice were anesthetized with xylazine-ketamine mix (Xilor 2% plus Imalgene 1000, Merial SpA, Milan, Italy), subjected to laparotomy and inoculated with 1 × 10^6^ luciferase (luc)-transfected SH-SY5Y (SH-SY5Y-luc) cell line into the left adrenal gland capsule, as previously described [53]. No mice died as a result of this procedure. Seven days post implantation, mice were randomly assigned to the different treatment groups, which received four intravenous injections of 1959-sss/DM3 (10 mg/kg) twice weekly. The tumour growth and the response to therapy were monitored by visualizing by BLI the luciferase activity associated to the tumor cells, as described [54].

In the systemic toxicity experiment, mice were anaesthetized with xylezine 24 h after the last day of treatments and blood was collected, through the retro-orbital sinus from each mouse, into anticoagulant-free tubes, for clinical chemistry hepatic, cardiac and renal evaluations. Samples were centrifuged at 2500× *g* for 10 min at 4 °C, and levels of serum albumin (ALB), phosphatase alkaline (ALP), glutamic-pyruvic transaminase (ALT), glutamic oxaloacetic transaminase (AST), lactate dehydrogenase (LDH), and creatinine (CREA) were quantified. All the reported evaluations were performed at the Mouse Clinic, IRCCS Ospedale San Raffaele, Milan.

#### 4.4.4. Biodistribution

1959-sss/DM3 accumulation in tumors was evaluated by immunofluorescence analysis of SKNAS or hNB tumor xenografts. Animals received a single injection of vehicle (PBS) or 1959-sss/DM3 intravenously and were sacrificed 72 h later. Fresh tumor tissues were frozen in a crio-embedding medium (OCT, BioOptica, Milan, Italy) and cryostat sections were incubated with the following antibodies: AlexaFluor-488 conjugated anti-human IgG 1:200 (A11013, Invitrogen, Life Technologies) in order to detect 1959-sss/DM3 and rat monoclonal anti-CD31 (550274, BD Pharmingen, Franklin Lakes, NJ, USA), mixed with rat monoclonal anti-CD105 (550546, BD Pharmingen) at the diluition of 1:40, followed by secondary antibody AlexaFluor-546 conjugated 1:200 (A11081, Molecular Probes, Life Technologies). Endothelial cells were identified by immunohistochemical labeling of tissue sections using a platelet-endothelial cell adhesion molecule (PECAM-1/CD31) and endoglin (CD105). Nuclei were stained with DRAQ5 (Alexis, Life Technologies, Carlsbad, CA, USA). Images acquisition was performed using Zeiss LSM 510 META confocal microscope. 

### 4.5. Expression Analysis of LGALS3BP 

#### 4.5.1. Western Blotting

Neuroblastoma cells (5 × 10^5^) were seeded in complete medium and after 48 h, they were lysed with RIPA buffer containing protease and phosphatase inhibitors (Sigma Aldrich Corporation, St. Louis; MO; USA). Lysates were clarified by centrifugation at 14,000× rpm for 10 min at 4 °C, subjected to 10% SDS-PAGE and Western blotting using a goat monoclonal antibody against LGALS3BP (AF2226, R&D Systems, Minneapolis, MN, USA) and a mouse human β-actin antibody (Sigma Aldrich Corporation). Incubation was performed overnight at 4 °C. After washing with PBS containing 0.1% Tween-20, blots were incubated with a goat anti-mouse HRP-conjugated IgG (Biorad, Berkeley, CA, USA) and an anti-goat HRP-conjugated antibody (Invitrogen, Life Technologies) as secondary antibodies, at room temperature for 1 h and developed with a chemiluminescence detection system (Biorad, Berkeley, CA, USA). Densitometry analysis was performed using ImageJ software, normalized relative to actin and expressed as arbitrary units. 

#### 4.5.2. ELISA

Neuroblastoma cells (5 × 10^5^) were seeded in complete medium and after 48 h, conditioned medium was removed and analyzed for secreted LGALS3BP by sandwich ELISA provided by Diesse Diagnostica Senese Spa (Siena, Italy), following manufacturing instructions. 

#### 4.5.3. Quantitative Reverse Transcription-PCR

Neuroblastoma cells (5 × 10^5^) were seeded in complete medium, after 48 h total RNA was isolated from cell lines by RNeasy mini kit (QIAGEN, Hilden, Germany). cDNAs were synthesized by Hyperscript First strand Synthesis Kit (GeneAll, Seoul, Korea) according to the manufacturers’ protocols. Relative mRNA expression analysis was performed by RT-qPCR into Bio-rad CFX96 Real-Time PCR Detection System (Applied Biosystems, Waltham, MA, USA), using SsoAdvanced Universal SYBR Green Supermix (Biorad) using the following primers: LGALS3BP (NM_005567.4) Fw 5′-GAACCCAAGGCGTGAACGAT-3′; Rev 5′-GTCCCACAGGTTGTCACACA-3′, and human β-actin Fw 5′-CAGCTCACCATGGATGATGATATC-3′ and Rev 5′-AAGCCGGCCTTGCACAT-3′ as housekeeping gene, and with an amplification program as follows: one cycle of 95 °C for 30 sec and 40 cycles of 95 °C for 15 s and 60 °C for 30 sec, followed by a melting curve according to the manufacturer’s protocol. All genes expression were normalized using human β-actin as housekeeping gene, and qRT-PCRs were performed in triplicate. Relative quantification of gene expression was calculated using the comparative cycle threshold (Ct) method of 2^−ΔCt^ [55].

### 4.6. Extracellular Vesicles (EVs) Purification and Analysis

Confluent SKNAS and hNB cells were cultivated for 48 h in serum-free medium. Around 100 mL of supernatant was collected and differential ultracentrifugation was performed for EVs isolation [56]. Briefly, supernatant was centrifuged at 300, 2000, 10,000 and 100,000× *g* for 10, 30 and 70 min, respectively, at 4 °C. The pellet of the last centrifugation consisted of crude extracellular vesicles, which was analyzed for LGALS3BP expression by Western Blotting and ELISA.

Lysis and western blotting of cells and EVs were performed as previously described, with some modifications. Briefly, whole cell lysate was lysed in RIPA lysis buffer containing protease and phosphatase inhibitors (Sigma Aldrich Corporation), while isolated EVs were prepared in a reducing or non-reducing (for CD9 immunoblotting, without 2-mercaptoethanol) sample buffer (50 mM Tris-HCl pH 6.8, 5% glycerol, 2% SDS, 1.5% 2-mercaptoethanol with bromophenol blue) and heated at 95 °C, 10 min. Lysates were electrophoresed and transferred to nitrocellulose; non-reduced blots were probed with primary antibody CD9 (sc-59140, Santa Cruz Biotechnology) and a mouse human β-actin antibody (Sigma Aldrich Corporation), both used in order to demonstrate the presence of EVs-associated protein; and with a goat monoclonal antibody against LGALS3BP (AF2226, R&D Systems), following by HRP-conjugated secondary antibodies. Blots were developed with a chemiluminescence detection system (Biorad, Berkeley, CA, USA).

Sandwich ELISA for LGALS3BP in intact EVs was done as follows: ninety-six well-plates NUNC were coated with murine anti-LGALS3BP antibody SP2 [57] (2 μg/mL) overnight at 4 °C. After blocking with 1% BSA in PBS for 1 h, 50 μL of intact EVs or PBS (as control) were added and incubated for 1 h at RT. After three washes with PBS-0.05% Tween-20, humanized anti-LGALS3BP antibody 1959 (1 μg/mL) was incubated for 1 h at RT. For the detection, after three washes with PBS-0.05% Tween-20, anti-human IgG-HRP (A0170, Sigma Aldrich) was added (1:5000) and incubated for 1 h at room temperature. After washes, stabilized chromogen was added for at least 10 min in the dark, before stopping the reaction with the addition of 1N H_2_SO_4_. The resulting color was read at 490 nm with Elisa plate reader. 

### 4.7. Confocal Imaging

For live cell staining, SKNAS or hNB cultured under standard growth conditions were plated at 70% of confluence on glass coverslips and after 24 h were incubated with 10 μg/mL of anti-LGALS3BP(1959) at 37 °C for 90 min in PBS and 3% of BSA. Afterwards the cells were washed in PBS, fixed in 4% paraformaldehyde, permeabilized and incubated with 1:200 AlexaFluor-488 conjugated anti-human IgG (A11013, Invitrogen, Life Technologies) and Hoechst 3342. Confocal images were acquired using a Zeiss LSM800 inverted confocal microscope system (Carl Zeiss, Gottingen, Germany). A single focal plane of the images was acquired under non-saturating conditions (pixel fluorescence below 255 arbitrary units) and using the same settings for all samples. 

For co-localization assays, A375m and SKNAS cells cultured under standard growth conditions were plated at 70% of confluence and were allowed to adhere on poly-L-lysine coated glass coverslips. After 24 h were fixed with 4% paraformaldehyde in PBS, for 10 min, at RT. The cells were incubated with 3% BSA 20 min at RT and, afterwards, were incubated with following antibodies: anti-CD 63 (1:50, Thermo Fisher Inc., Waltham, MA, USA), anti-CD81 (1:50, Thermo Fisher Inc.) and anti-Gal-3BP (1:500) at 4 °C O.N. Primary antibodies were revealed by AlexaFluor-633 conjugated anti-human IgG (1:2000, Invitrogen, Life Technologies) and AlexaFluor-488 conjugated anti-mouse IgG (1:2000 Invitrogen, Life Technologies). Coverslips were mounted with Vectashield Mounting Medium with Dapi (Vector Laboratories, Burlingame, CA, USA) and examined at a Leica TCS SP5 confocal microscope (Mannheim, Germany).

LGALS3BP/CD63 and CD81 co-localization was quantified by Image J and Pearsons’s correlation index was calculated as previously described [58]. 

### 4.8. Immunohistochemistry 

Paraffin-embedded blocks were prepared from xenograft specimens after being fixed with 10% neutral buffered formalin. 4μm-thick sections were immunostained for LGALS3BP using the 1A4.22 monoclonal antibody [31]. Microwave pretreatment (10 min) in citrate buffer (pH 6.0) was performed for antigen retrieval. Endogenous biotin was saturated with a biotin blocking kit (Vector Laboratories, Burlingame, CA, USA). The Vectastain ABC peroxidase kit (Vector Laboratories) was used to detect the antigen. The 3,3-diaminobenzidine (DAB; Agilent, Santa Clara, CA, USA) was used as chromogen. Finally, the sections were counterstained with Mayer’s hematoxylin, dehydrated, cleared, and coverslipped for light microscopic examination. Negative controls were obtained using a matched isotype control antibody. 

### 4.9. Statistical Analysis 

For metastasis analysis, P values were determined by Student’s t test and considered significant for *p* < 0.05. For Kaplan Meier survival analysis, a Log-rank (Mantel-Cox) test was used to compare each of the arms. Experimental sample numbers (n) are indicated in the Figures. All statistical analysis was performed with GraphPad Prism 5.0 software.

## 5. Conclusions

Neuroblastoma is a solid childhood cancer that still poses a great challenge to clinicians. Immunotherapy in the context of this neoplasm is limited to the use of a naked antibody targeting the GD2 antigen, which can extend survival of a few patients. The results presented in this investigation demonstrate that a newly developed ADC targeting LGALS3BP has the potential to eradicate metastatic and MYCN amplified neuroblastomas in preclinical mouse models. If validated by further safety studies, 1959-sss/DM3 could be added to the arsenal of immunotherapeutics for treatment of neuroblastoma and, potentially, other malignancies expressing LGALS3BP.

## Figures and Tables

**Figure 1 cancers-12-02989-f001:**
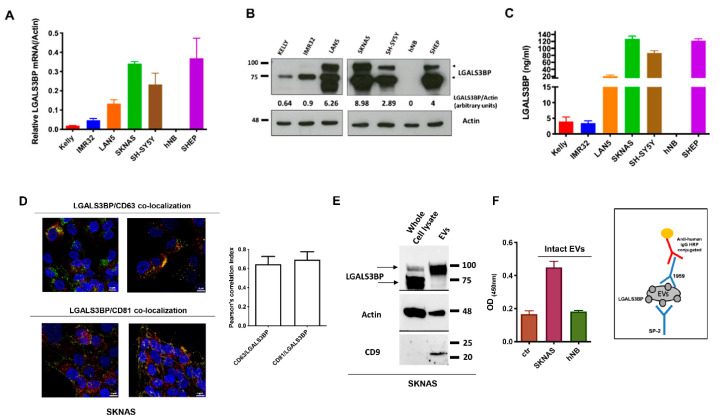
LGALS3BP is expressed and secreted in neuroblastoma. (**A**–**C**) LGALS3BP mRNA, intracellular and secreted protein levels in a panel of human neuroblastoma cell lines were analyzed using quantitative RT-PCR, WB and ELISA, respectively. Band intensities (**B**) were quantified and indicated in figure as arbitrary units. (**D**) Confocal imaging showing LGALS3BP (red) co-localization with CD63 and CD81 (green) in SKNAS neuroblastoma cells. Cells nuclei were stained by DRAQ5 (blue). Images taken at 63X magnification. Scale bar: 10 μM. (**E**) Immunoblot showing LGALS3BP and the exosomal markers Actin and CD9 expression in EVs isolated from SKNAS cell culture supernatant compared to whole cell lysate. (**F**) Sandwich ELISA performed on intact EVs isolated from (LGALS3BP+) SKNAS and (LGALS3BP-) hNB supernatants (left panel). Scheme of sandwich ELISA is illustrated (right panel). Ctr: ELISA blank control. Arrows (**B**,**E**) indicate the intracellular (70 kDa) and the fully mature (90 kDa) forms of LGALS3BP.

**Figure 2 cancers-12-02989-f002:**
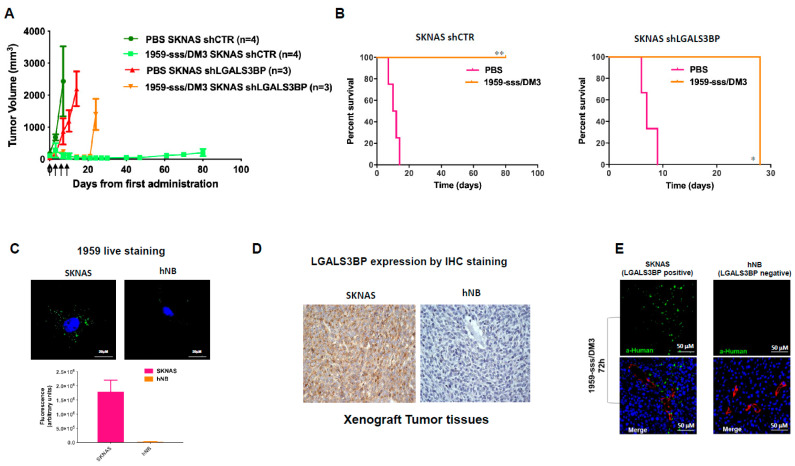
1959-sss/DM3 induces tumor shrinkage in a target-dependent manner. (**A**) CD1 nude mice harboring silenced shLGALS3BP or control vector-infected (shCTR) SKNAS xenografts were treated with vehicle (PBS) or 1959-sss/DM3 at the dose of 10 mg/kg twice weekly. (**B**) Kaplan-Meier survival curves are shown. Log-rank (Mantel-Cox) Test. * *p* = 0.02; ** *p* = 0.006. (**C**) Confocal images of live neuroblastoma cells labelled with humanized 1959 anti-LGALS3BP antibody for 90 min at 37 °C followed by AlexaFluor 488 conjugated secondary anti-human IgG antibody (green). Cells nuclei were stained by DRAQ5 (blue). Quantification of total immunofluorescence staining per cell is shown in arbitrary units. (**D**) IHC staining of neuroblastoma xenografts with 1A4.22 anti-LGALS3BP antibody. (**E**) 1959-sss/DM3 in vivo accumulation (green) in LGALS3BP positive (SKNAS) and negative (hNB) neuroblastoma xenografts. Blood vessels were stained using anti CD31/CD105 antibodies (red); cells nuclei were stained by DRAQ5 (blue).

**Figure 3 cancers-12-02989-f003:**
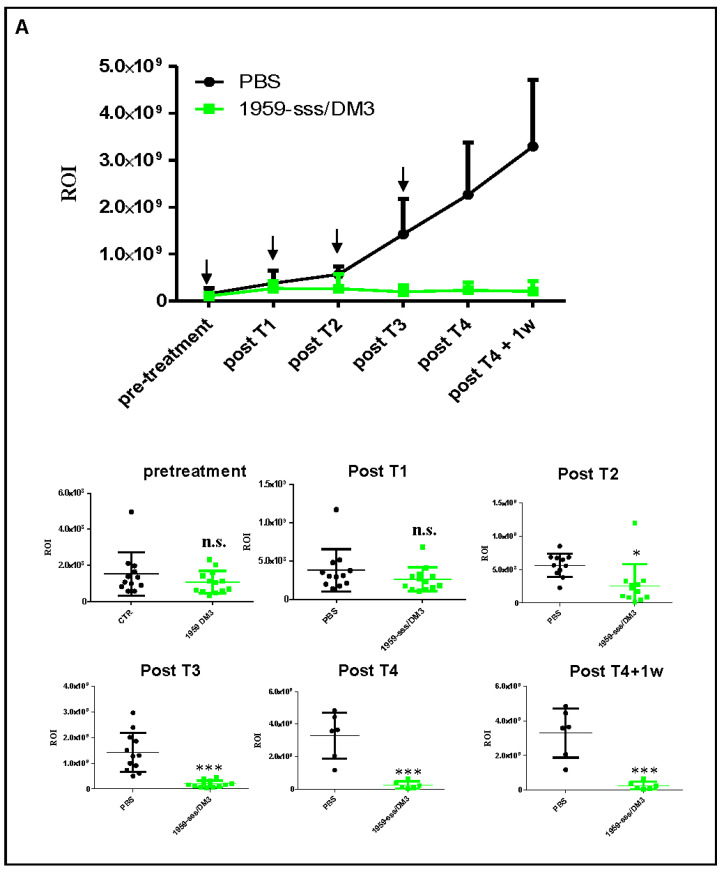
Therapeutic activity of 1959-sss/DM3 in SH-SY5Y-LUC orthotopic xenograft model. (**A**) Orthotopic SH-SY5Y-LUC neuroblastoma xenograft showing potent antitumor activity of intravenous 1959-sss/DM3 administration (arrows) at the dose of 10 mg/kg twice weekly. Growth curve obtained was expressed as tumor region of interest (ROI). In the below panels, single animals ROI are shown. Unpaired T Test. * *p* = 0.01; *** *p* < 0.0001. (**B**) Tumor weight of animals sacrificed 24 h after the last treatment. Unpaired T Test. * *p* = 0.03. (**C**) Survival curves evaluated by Kaplan-Meier and analyzed by Log-rank (Mantel-Cox). *** *p* = 0.0001. n.s. (not significant).

**Figure 4 cancers-12-02989-f004:**
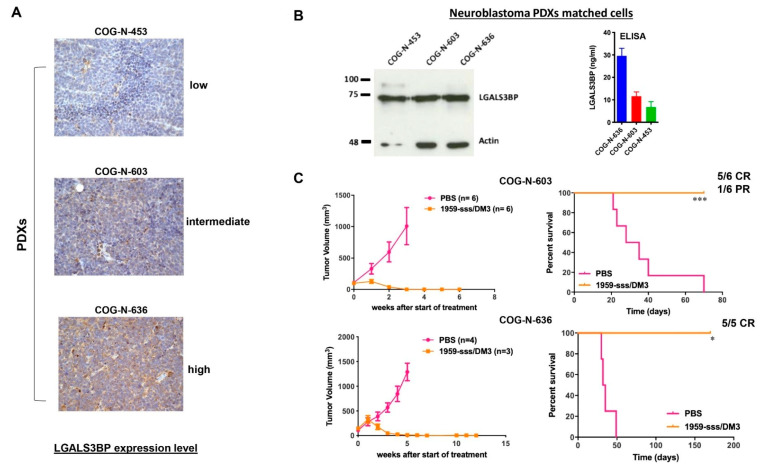
1959-sss/DM3 induces tumor shrinkage in neuroblastoma PDX models. (**A**) LGALS3BP expression was evaluated by IHC in three neuroblastoma PDXs by IHC and in matched cell lines (**B**) by WB and ELISA. (**C**) Subcutaneous COG-N-603 and COG-N-636 PDXs growth curves of intravenous 1959-sss/DM3 administration at the dose of 10 mg/kg twice weekly for 4 doses. Survival curves were evaluated by Kaplan-Meier and analyzed by the log-rank test using Graphpad Prism 5 software. * *p* = 0.01; *** *p* = 0.0005. CR: complete remission; PR: partial response.

**Figure 5 cancers-12-02989-f005:**
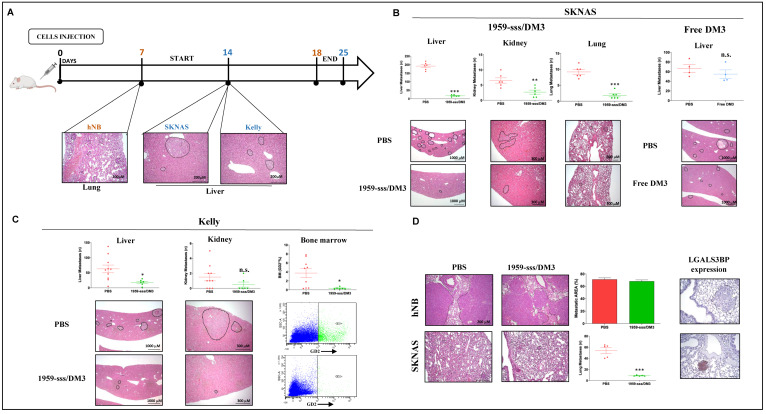
Targeting LGALS3BP by 1959-sss/DM3 inhibits metastatic spreading. (**A**) Schematic representation of experimental metastasis assays performed in SKNAS (**B**), Kelly (**C**) and hNB (**D**) neuroblastoma cell models. PBS, 1959-sss/DM3 (10 mg/kg), or free SH-DM3 (0.1 mg/kg) intravenous injections were performed every three days for a total of 3 doses. Metastatic lesions were analyzed and plotted on graphs as number of lesions (SKNAS and Kelly) or as percentage of metastatic area (hNB). Representative images of H&E staining are shown below. Bone marrow metastasis (**C**) were analyzed by FACS analysis and plotted on graphs as percentage of GD2+ cells. Representative dot plots are shown below. Expression of LGALS3BP on lung metastatic lesions was evaluated by IHC staining (**D**). Unpaired T Test. * *p* = 0.01; ** *p* = 0.006; *** *p* < 0.0001. n.s. (not significant).

## Data Availability

All data generated or analyzed during this study are included in this published article.

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
