# Peer review of "Targeting Vesicular LGALS3BP by an Antibody-Drug Conjugate as Novel Therapeutic Strategy for Neuroblastoma"

_cancers, 2020, doi:10.3390/cancers12102989_

Round 1

Reviewer 1 Report

The authors present a novel and potential treatment for neuroblastoma patients with intermediate and high expression level of LGALS3BP by using a humanized non-internalizing ADC 1959-sss/DM3. This is very interesting, and I would be happy to see this treatment can go further. I have some minor points for this paper, please see below:

  1. Line 29: a space between ‘LGALS3BP’ and ‘is’.
  2. Please provide a description about the conjugated drug SH-DM3. Not all the readers know how it works, so a brief description will make it easier to read through. And explain a little bit in the ‘Introduction’ how this non-internalizing ADC can kill tumor cells, though you provided several references.
  3. In this paper it was mentioned that neuroblastoma-derived LGALS3BP-containing EVs can be incorporated by mesenchymal stromal cells, so it would be interesting to know whether this ADC can be internalized by MSCs.
  4. Regarding suppl. Fig 1, is it possible to include some MSCs like cancer-associated fibroblast and macrophages to your setting?
  5. Line 80-82: it would be interesting to know a little bit about the genetic background of this hNB cell line, like whether MYCN amplified, ALK mutant or 11q-deleted.
  6. Figure 1D: please do this using a neuroblastoma cell line, instead of this A375m melanoma cell line, since this paper is about the usage of LGALS3BP ADC in neuroblastoma.
  7. Figure 1B: concerning the western blot bands of LGALS3BP, why in some cell lines (like Kelly and IMR32) only 1 band and why in some cells there are 2 or 3 bands? Is it caused by expression level, or other reasons?
  8. The order for figures 4 and 5 was wrong, please change them back.
  9. Figure 4: the IHC showed the lowest expression of LGALS3BP for PDX COG-N-453, but this is not consistent with the western blot result. If you normalize with actin control, you can see actually it has the highest expression level.
  10. While after treatment 5 out of 6 mice bearing the COG-N-603 PDX were disease-free, what happened to the remaining one?
  11. Suppl. Figure2: 2B, it seems the treatment with 1959-sss/DM3 affects the mouse body weight a little bit, but no information to see whether this is statistically significant. Also, for the biochemical analysis of blood samples, I noticed that the sample numbers of PBS group and 1959-sss/DM3 group were not consistent among these tests, why? And, the decreased ALB level and increased ALP level as shown in the figure, are they significant enough to say this ADC causes certain liver damage?
  12. Discussion line 224-225: I don’t agree with the authors here. Though the treatment of ALK-driven neuroblastoma patients with 1st generation of ALK inhibitor crizotinib was less promising due to the intrinsic resistance of ALK point mutations to this drug, second and third generations ALK inhibitors were developed to overcome this resistance and showed great therapeutical efficacy. Papers also show these next-generation ALK inhibitors were able to cure the patient successfully.
  13. See references for point 12: 1) Mossé YP, Lim MS, Voss SD, et al. Safety and activity of crizotinib for paediatric patients with refractory solid tumours or anaplastic large-cell lymphoma: a Children's Oncology Group phase 1 consortium study. Lancet Oncol. 2013;14(6):472-480. doi:10.1016/S1470-2045(13)70095-0; 2) Infarinato NR, Park JH, Krytska K, et al. The ALK/ROS1 Inhibitor PF-06463922 Overcomes Primary Resistance to Crizotinib in ALK-Driven Neuroblastoma. Cancer Discov. 2016;6(1):96-107. doi:10.1158/2159-8290.CD-15-1056; 3) Guan J, Tucker ER, Wan H, et al. The ALK inhibitor PF-06463922 is effective as a single agent in neuroblastoma driven by expression of ALK and MYCN. Dis Model Mech. 2016;9(9):941-952. doi:10.1242/dmm.024448; 4) Guan J, Fransson S, Siaw JT, et al. Clinical response of the novel activating ALK-I1171T mutation in neuroblastoma to the ALK inhibitor ceritinib. Cold Spring Harb Mol Case Stud. 2018;4(4):a002550. Published 2018 Aug 1. doi:10.1101/mcs.a002550
  14. It would be good to add some information about whether the increased serum level of LGALS3BP affects the efficacy of this ADC, and whether this ADC affects blood cells.
  15. As mentioned that the disulfide-linked payload is released within the reducing tumor microenvironment, so inside the body are there any other organs/tissues which can provide this reducing environment? If there are, then how this ADC will affect these organs/tissues, though the majority might be enriched in the tumor tissue.

Author Response

Dear Editor,

We thank the referee for her/his constructive criticisms and suggestions, which allowed us to significantly improve the manuscript. We hope the manuscript can be accepted in this revised form.

Changes responding to ref#1 are yellow highlighted in the revised text.

Please find below our point-by-point answers:

Ref  #1

  1. Line 29: a space between ‘LGALS3BP’ and ‘is’.

Corrected.

  1. Please provide a description about the conjugated drug SH-DM3. Not all the readers know how it works, so a brief description will make it easier to read through. And explain a little bit in the ‘Introduction’ how this non-internalizing ADC can kill tumor cells, though you provided several references.

We have added the following sentence in the Introduction:

“DM3 is a chemical derivative of maytansine belonging to the tubulin-binding ADC payload class with a cell killing potency in the picomolar range  [1]“

In addition, we have provided a specific reference for this type of drug [1]

  1. In this paper it was mentioned that neuroblastoma-derived LGALS3BP-containing EVs can be incorporated by mesenchymal stromal cells, so it would be interesting to know whether this ADC can be internalized by MSCs.

We agree with referee regarding this intriguing observation. Studies are planned to address in vitro and possibly on animal models the hypothesis whether 1959/ADC may be internalized by neuroblastoma-associated stromal cells, such as tumor-activated macrophages and fibroblasts.

  1. Regarding suppl. Fig 1, is it possible to include some MSCs like cancer-associated fibroblast and macrophages to your setting?

According to the reviewer suggestion we have now analyzed and reported SH-DM3 cell killing activity in RAW-264.7 macrophages upon stimulation with conditioned medium obtained by NB SKNAS cells, as previously reported [2]. These new data are presented in the new Supplem Fig 2B of the revised manuscript.

  1. Line 80-82: it would be interesting to know a little bit about the genetic background of this hNB cell line, like whether MYCN amplified, ALK mutant or 11q-deleted.

Supplemental information about hNB cell line have been added in the Methods section

“The primary human neuroblastoma HNB cell line was isolated from a non-MYCN amplified tumour metastasised in the neck of a 3-year-old male patient. The 11q and ALK status is not known.”

  1. Figure 1D: please do this using a neuroblastoma cell line, instead of this A375m melanoma cell line, since this paper is about the usage of LGALS3BP ADC in neuroblastoma.

As suggested, we have now included LGALS3BP/EVs markers co-localization in Neuroblastoma SKNAS cells. These new data are now in the Figure 1D of the revised manuscript. We moved A375m melanoma images to a new Supplemental Figure 1.

  1. Figure 1B: concerning the western blot bands of LGALS3BP, why in some cell lines (like Kelly and IMR32) only 1 band and why in some cells there are 2 or 3 bands? Is it caused by expression level, or other reasons?

As for Western Blot bands shown in Figure 1B, we would like to point out that LGALS3BP is a hyperglycosilated protein. The glycosylation pattern may differ within cell lines. Two main intracellular isoforms can be detected by WB (70-90 KDa), reflecting the presence of cytoplasmic (70KDa) and mature, ready to be secreted (90Kda) forms. Different isoforms are now indicated in Figure 1 with arrows.

  1. The order for figures 4 and 5 was wrong, please change them back.

            We are sorry for the mistake. Figures order is now corrected.

  1. Figure 4: the IHC showed the lowest expression of LGALS3BP for PDX COG-N-453, but this is not consistent with the western blot result. If you normalize with actin control, you can see actually it has the highest expression level

While the intracellular level of LGALS3BP (i.e. target expression) is similar among the three PDX models,  we would like to underline that the amount of secreted protein  (i.e. the target of the ADC) resulted to be higher (ELISA and IHC staining) in PDX COG-N-603 and PDX COG-N-636, and this is the reason why these tumors were selected for the in vivo studies.

To better clarify this point in the manuscript we have now modified the sentence in the Results section as follows:

Three highly aggressive, stage IV, MYCN amplified PDXs,  COG-N-636, COG-N-603 and COG-N-453, were used. COG-N-636 and COG-N-603, which overall showed high expression levels of ADC target were selected for the in vivo studies (Figure 4A-B).

  1. While after treatment 5 out of 6 mice bearing the COG-N-603 PDX were disease-free, what happened to the remaining one?

We thanks the referee for this comment. At the time of submission, five out of six animals were disease free while the remaining one showed relapse, thus indicating for this sample a Partial Response instead of Complete Remission. We have now modified, results section, figure and the legend accordingly.

To better clarify this point we have rephrased the sentences in the results section as following:

“Treatments with the 1959-sss/DM3 ADC caused complete eradication of tumours in the COG-N-636 PDX model, whereas 5 out of 6 mice injected with COG-N-603 PDX cells were tumour-free at the time of manuscript submission, with only one animal showing a partial response (Figure 4C). The therapeutic activity was long-lasting, since tumour regrowth was only observed in one of the COG-N-603 PDX transplanted mice after 70 days from antibody injections. None of the COG-N-636 PDX transplanted mice showed signs of tumour regrowth after 150 days since the start of treatments.”

  1. Suppl. Figure2: 2B, it seems the treatment with 1959-sss/DM3 affects the mouse body weight a little bit, but no information to see whether this is statistically significant. Also, for the biochemical analysis of blood samples, I noticed that the sample numbers of PBS group and 1959-sss/DM3 group were not consistent among these tests, why? And, the decreased ALB level and increased ALP level as shown in the figure, are they significant enough to say this ADC causes certain liver damage?

Mice body weight is reported in Suppl. Figure 2. The difference in the body weight of animals treated with ADC versus PBS is  not statistically  significant. In the biochemical analysis of blood samples the referee correctly noticed that there’s no consistency between tests and animal samples. This was due to the low amount of available blood samples for some animals. We have reported this information in the figure legend.

 Decreased ALB and increased ALP levels in ADC-treated mice are not significantly different compared to those observed in vehicle only- injected animals. Moreover, AST and ALT levels appear stable, and therefore we would exclude liver damage.

  1. Discussion line 224-225: I don’t agree with the authors here. Though the treatment of ALK-driven neuroblastoma patients with 1st generation of ALK inhibitor crizotinib was less promising due to the intrinsic resistance of ALK point mutations to this drug, second and third generations ALK inhibitors were developed to overcome this resistance and showed great therapeutical efficacy. Papers also show these next-generation ALK inhibitors were able to cure the patient successfully.

We agree with the point raised by the referee. We have modified the sentence in the Discussion as follows:

However, the main neuroblastoma oncogene, MYCN, is still undruggable, and small molecule inhibitors targeting kinases activated in neuroblastoma, such PI3K or first generation ALK inhibitors, are either too toxic or made futile by the development of drug resistance, requiring the continuous search for more effective inhibitors [37, 38]. Although,  encouraging results were recently obtained with second and third generation ALK inhibitors which showed to overcome resistance and demonstrated significant therapeutic efficacy in several studies [3-6] .

See references for point 12: 1) Mossé YP, Lim MS, Voss SD, et al. Safety and activity of crizotinib for paediatric patients with refractory solid tumours or anaplastic large-cell lymphoma: a Children's Oncology Group phase 1 consortium study. Lancet Oncol. 2013;14(6):472-480. doi:10.1016/S1470-2045(13)70095-0; 2) Infarinato NR, Park JH, Krytska K, et al. The ALK/ROS1 Inhibitor PF-06463922 Overcomes Primary Resistance to Crizotinib in ALK-Driven Neuroblastoma. Cancer Discov. 2016;6(1):96-107. doi:10.1158/2159-8290.CD-15-1056; 3) Guan J, Tucker ER, Wan H, et al. The ALK inhibitor PF-06463922 is effective as a single agent in neuroblastoma driven by expression of ALK and MYCN. Dis Model Mech. 2016;9(9):941-952. doi:10.1242/dmm.024448; 4) Guan J, Fransson S, Siaw JT, et al. Clinical response of the novel activating ALK-I1171T mutation in neuroblastoma to the ALK inhibitor ceritinib. Cold Spring Harb Mol Case Stud. 2018;4(4):a002550. Published 2018 Aug 1. doi:10.1101/mcs.a002550

We have now added these references as suggested.

  1. It would be good to add some information about whether the increased serum level of LGALS3BP affects the efficacy of this ADC, and whether this ADC affects blood cells
  2. As mentioned that the disulfide-linked payload is released within the reducing tumor microenvironment, so inside the body are there any other organs/tissues which can provide this reducing environment? If there are, then how this ADC will affect these organs/tissues, though the majority might be enriched in the tumor tissue.

 We thanks the referee for these pertinent observations. In our previous paper [7], we performed a preliminary toxicity study in rabbits, which is the only species in where 1959  antibody cross reacts with the endogenous LGALS3BP. In this study, we did not observe any hematological and biochemical alteration, thus suggesting that the ADC is well tolerated in the presence of the endogenous protein.

The analysis of ADC efficacy in the presence of LGALS3BP serum level is a well taken point. Because of loss of cross-reactivity with the mouse protein, this aspect cannot be addressed in xenograft experiment in mice. The only model to study the efficacy of the ADC in the presence of the target in serum is the rabbit. We are planning to perform this kind of experiment in rabbits. 

We have added the following sentence in the discussion:

Moreover, therapeutic studies using a rabbit tumor model [8] should be performed to predict whether the serum level of LGALS3BP may affect the efficacy of this ADC due to a potential sink effect.

References

  1. Zhang, D., et al., Exposure-Efficacy Analysis of Antibody-Drug Conjugates Delivering an Excessive Level of Payload to Tissues. Drug Metab Dispos, 2019. 47(10): p. 1146-1155.
  2. Hashimoto, O., et al., Collaboration of cancer-associated fibroblasts and tumour-associated macrophages for neuroblastoma development. J Pathol, 2016. 240(2): p. 211-23.
  3. Guan, J., et al., Clinical response of the novel activating ALK-I1171T mutation in neuroblastoma to the ALK inhibitor ceritinib. Cold Spring Harb Mol Case Stud, 2018. 4(4).
  4. Infarinato, N.R., et al., The ALK/ROS1 Inhibitor PF-06463922 Overcomes Primary Resistance to Crizotinib in ALK-Driven Neuroblastoma. Cancer Discov, 2016. 6(1): p. 96-107.
  5. Guan, J., et al., The ALK inhibitor PF-06463922 is effective as a single agent in neuroblastoma driven by expression of ALK and MYCN. Dis Model Mech, 2016. 9(9): p. 941-52.
  6. Mosse, Y.P., et al., Safety and activity of crizotinib for paediatric patients with refractory solid tumours or anaplastic large-cell lymphoma: a Children's Oncology Group phase 1 consortium study. Lancet Oncol, 2013. 14(6): p. 472-80.
  7. Giansanti, F., et al., Secreted Gal-3BP is a novel promising target for non-internalizing Antibody-Drug Conjugates. J Control Release, 2018. 294: p. 176-184.
  8. Aravalli, R.N. and E.N. Cressman, Relevance of Rabbit VX2 Tumor Model for Studies on Human Hepatocellular Carcinoma: A MicroRNA-Based Study. J Clin Med, 2015. 4(12): p. 1989-97.

Reviewer 2 Report

This is an interesting manuscript focusing on development of a non-internalising Antibody-Drug Conjugates (ADC)s-based therapy for neuroblastoma. Study is extending from author’s earlier report on a similar approach, which was demonstrated using a melanoma model (Giansanti et al., 2019). In this manuscript using different types of neuroblastoma models Capone et al describing that 1959-sss/DM3 selectively targets LGALS3BP expressing neuroblastoma and mediate therapeutic activity. Manuscript is generally well written and arguments supported with data, I have several minor comments regarding this manuscript.

Comments:

Figure 1B (Western blot) needs quantification

Figure 1D Since the study is focusing on neuroblastoma models authors should show co-localasiation of LGALS3BP with EV CD63 and CD81 in selected neuroblastoma cells (eg SKNAS).

Figure 1F An additional control is needed here. Authors could including EVs from hNB cells as negative control to demonstrate the signal is specific.

Suppl Figure 1A SKNAS cell model appears less sensitive to treatment, authors should comment on this. OD for hNB cells >3, for most equipment data obtained at this high OD will not be linear. Methodology for MTT assay is missing.

Figure 2 Authors should comment on/ justify selecting SKNAS cell line for further analysis (since several neurobalstoma cell lines showed high expression of the target-Figure 1).

Figure 4 and Figure 5 swapped in order in manuscript (Figure legends are not for the figures presented).

Discussion: Considering neuroblastoma is a heterogeneous cancer, with many different clinical presentations (which is also evidenced by authors own data that not all neuroblastomas express LGALS3BP, and the expression levels vary greatly within tested cell lines) authors should acknowledge potential limitations of this approach/target.  

Author Response

Dear Editor,

We thank the referee for her/his constructive criticisms and suggestions, which allowed us to significantly improve the manuscript. We hope the manuscript can be accepted in this revised form.

Changes relative to ref#2 are turqoise highlighted in the revised text.

Please find below our point-by-point answers:

Ref#2

Figure 1B (Western blot) needs quantification

In the revised manuscript Fig1B reports the densitometry of the WB.

Figure 1D Since the study is focusing on neuroblastoma models authors should show co-localasiation of LGALS3BP with EV CD63 and CD81 in selected neuroblastoma cells (eg SKNAS).

We have now included Gal3BP/EV co-localization in NB Sknas cells, these new data are now in the Figure 1D of the revised manuscript. We moved A375 imaging data on Supplemental Figure 1.

Figure 1F An additional control is needed here. Authors could including EVs from hNB cells as negative control to demonstrate the signal is specific.

We agree with the referee observation. We have now included WB and ELISA on hNB as negative control (Figure 1F and Supplemental Figure 1 of the revised manuscript).

Suppl Figure 1A SKNAS cell model appears less sensitive to treatment, authors should comment on this. OD for hNB cells >3, for most equipment data obtained at this high OD will not be linear. Methodology for MTT assay is missing.

Figure 2 Authors should comment on/ justify selecting SKNAS cell line for further analysis (since several neurobalstoma cell lines showed high expression of the target-Figure 1).

We have implemented Methods section indicating methodology used for the MTT assays. We have performed new assays on hNB cells in a OD linear range.  We agree with the referee that Sknas show less in vitro sensitivity to DM3 payload, although  these cells do express and secrete very high level of LGALS3BP.

In order to analyze therapeutic activity of the ADC we have chosen three different available cell lines known to be able to form tumor in immunodeficient mice, i.e. SKNAS, SHSY-5Y and Kelly. Moreover, therapeutic activity was confirmed in patient derived xenografts (PDX). 

Figure 4 and Figure 5 swapped in order in manuscript (Figure legends are not for the figures presented).

5) The order of the Figures has been corrected

Discussion: Considering neuroblastoma is a heterogeneous cancer, with many different clinical presentations (which is also evidenced by authors own data that not all neuroblastomas express LGALS3BP, and the expression levels vary greatly within tested cell lines) authors should acknowledge potential limitations of this approach/target. 

6) We add the following sentence in the Discussion section:

“It is important to underline that neuroblastoma, similar to other cancers, is a very heterogeneous tumor, and therefore the different levels of expression of the target in the patient population (as observed in our PDX models) could make this therapeutic approach applicable only in a selected group of neuroblastoma patients”